# Increased Barefoot Stride Variability Might Be Predictor Rather than Risk Factor for Overuse Injury in the Military

**DOI:** 10.3390/ijerph20156449

**Published:** 2023-07-26

**Authors:** Darja Nesterovica-Petrikova, Normunds Vaivads, Ainārs Stepens

**Affiliations:** 1Military Medicine Research and Study Center, Rīga Stradiņš University, LV-1048 Riga, Latvia; ainars.stepens@rsu.lv; 2Joint Headquarters of the Latvian National Armed Forces Medical Service, LV-1006 Riga, Latvia

**Keywords:** musculoskeletal injuries, military personnel, gait analysis, stride variability, infantry boot

## Abstract

Footwear usage could be a promising focus in reducing musculoskeletal injury risk in lower extremities commonly observed among the military. The goal of this research was to find potential gait-related risk factors for lower leg overuse injuries. Cases (*n* = 32) were active-duty infantry soldiers who had suffered an overuse injury in the previous six months of service before enrolling in the study. The control group (*n* = 32) included infantry soldiers of the same age and gender who did not have a history of lower leg overuse injury. In the gait laboratory, individuals were asked to walk on a 5-m walkway. Rearfoot eversion, ankle plantar/dorsiflexion and stride parameters were evaluated for barefoot and shod conditions. Barefoot walking was associated with higher stride time variability among cases. According to the conditional regression analysis, stride time variability greater than 1.95% (AUC = 0.77, 95% CI (0.648 to 0.883), *p* < 0.001) during barefoot gait could predict lower leg overuse injury. Increased barefoot gait variability should be considered as a possible predictive factor for lower leg overuse injury in the military, and gait with military boots masked stride-related differences between soldiers with and without lower leg overuse injury.

## 1. Introduction

Military service is physically challenging and requires high volumes of marching and running activities. Non-combat musculoskeletal overuse injuries of the lower extremities have military readiness and socioeconomic impacts [1,2,3,4]. Despite years of musculoskeletal injury research and the injury risk prevention strategies implemented in the military, the prevalence of lower leg overuse injuries remains high. The reported prevalence of lower leg overuse injuries vary from 8% in the foot, to 22% in the knee region, and 34% in the calf and ankle [5]. Reduction in physical activity volume, bracing for high-risk activities, high level of pre-accession physical fitness, and awareness of injury prevention strategies are reported to reduce injury rates in military populations [6]. According to a recent meta-analysis, the evidence base of musculoskeletal injury preventive strategies remains insufficient to provide strong recommendations for practice [7].

Several risk factors for lower extremity overuse injury have been previously identified, including age, gender and peak plantar pressure [8,9]. Jacobsson et al. discovered an elevated risk of sustaining a subsequent injury in athletes with inadequate primary injury recovery [10]. In the military and athletic populations, previous injury increased the risk of subsequent lower limb injuries, and altered gait biomechanics [9,11,12,13].

Gait is a functional rhythmical movement, and complex fluctuations of unknown origin appear in the normal pattern among healthy individuals [14,15]. Although significant variation of gait is most often observed in movement disorders [16], few studies have looked at changes in the coefficient of variation of gait parameters (variability) as a risk factor or as a result of an injury [17]. Psoriatic arthritis patients showed higher stride variability [18], while patients with traumatic brain injury had increased step width variability [19]. Adults with a history of musculoskeletal injury had higher running gait variability than injury-free individuals [20,21].

Footwear usage impacts physical task performance and acutely affects gait stability and variability [22,23,24]. During service time, military personnel use military boots and running shoes, and shoe usage differs between countries and services [9]. The role of military footwear in protecting military personnel against musculoskeletal injuries of the lower extremities remains debatable. Considering the impact of footwear on gait biomechanics, the goal of this study was to identify a gait-related predictor for lower leg overuse injury in previously injured and non-injured infantry soldiers while walking barefoot and in military boots.

## 2. Materials and Methods

### 2.1. Study Population

Case-control study subjects were active-duty infantry male soldiers from the Latvian Land Forces. Subjects were selected after a cross-sectional study regarding the status of musculoskeletal injury. Injuries that occurred due to repetitive or forceful tasks and resulted from repeated overstretching or overloading were defined as overuse musculoskeletal injuries [3,25]. Common lower leg overuse injuries in the military are anterior or posterior tibial syndrome (ICD-10 code M76.8) [26], plantar fasciitis (M72.2), Achilles tendonitis (or bursitis, M76.6), peroneal tendinitis (M76.7), and stress fractures (M84.3) [9,27]. A detailed selection process was presented in a previous article [28]. Cases (*n* = 32) were subjects with a history of overuse injury in the lower leg, ankle, or foot during the last 6 months of service before entering the study. The injury was considered if a soldier had a medical record or reported an injury that restricted at least one activity. The recovery time from a musculoskeletal overuse injury ranged between 3 and 12 weeks [29,30], and the study started from 4 to 6 months after the injury occurred. The cases experienced complete recovery from injury before the research period, did not have functional limitations, and could participate in all kinds of physical activities. Controls were age and gender-matched infantry soldiers (*n* = 32) with no history of lower leg overuse injury. The characteristics of the study population are shown in Table 1. Participation was voluntary, and all study subjects provided written informed consent before entering the study. The study protocol was approved by the Ethics Committee of Rīga Stradiņš University (No. 40/26.10.2017).

### 2.2. Gait Assessment

Subjects in shorts were advised to walk comfortably barefoot on a straight 5-m walkway (active area). Shod walking trials were used to evaluate the effects of military boots on gait. During the shod gait analysis, a standardized infantry boot model for hot weather conditions with a 25 cm height was used; the worn boots had no visible attrition signs. Two gait trials were used for familiarization [31] with each gait condition (barefoot, shod) and were not investigated for reliable gait parameter measurement. Walking trials continued until full *n* = 50 gait cycles were recorded, and only straight walking patterns were included in the research to access gait variability; spatiotemporal stride parameters before/after turns were not evaluated [32,33]. 

All study subjects were fitted with spherical retroreflective markers (*n* = 12) using double-sided tape for motion tracking and gait cycle analysis. A single examiner placed markers bilaterally on the subject’s anatomical landmarks of the foot and shank. Markers were attached to the middle shank, the lateral and medial femoral epicondyles, the lateral and medial malleoli, the heads of the first, second, and fifth metatarsals, and the posterior calcaneus. During the shod condition, markers were placed after palpation of the anatomical landmark through the shoe. The marker set used in this study is similar to the conventional lower-limb gait model marker set (*n* = 8) and showed good test–retest reliability (ICC > 0.80) [34]. The markers were placed in the same locations as in previous studies for barefoot and shod conditions [35,36]. 

The study was carried out in the Riga Stradiņš University gait laboratory, which was equipped with two high-speed camera motion capture systems (100 samples/s) for video recording of gait. Data from marker tracking and Quintic v31 Biomechanics software (Quintic Consultancy Ltd., United Kingdom) were used to analyze 2D kinematics and spatiotemporal gait parameters [37,38]. Rearfoot eversion and ankle plantar/dorsiflexion angles were measured throughout the gait cycle’s stance phase. Heel contact was defined as the initial contact [39]. The foot contact angle was defined as the angle created between the foot and the ground during a heel strike. The anteroposterior distance between the left and right heel markers at each initial contact was used to calculate the step length. The definitions of spatiotemporal gait parameters presented in this study were the same as in a previous study that investigated lower-limb overuse injuries among military recruits [40]. The stride time variability was calculated as 100 × (stride time SD/mean stride time), the stride length variability was calculated as 100 × (stride length SD/mean stride length), and the step length asymmetry was calculated as 100 × ln (right step length/left step length).

### 2.3. Data Processing and Statistical Analysis

Sample size was calculated using the open-source calculator (OpenEpi, Open Source Statistics for Public Health) [41], estimate based on one-year musculoskeletal lower extremity injury among Latvian Land Forces (12.4%) [8].

Statistical analysis was performed using the SPSS 22.0 software package (Statistical Package for the Social Sciences). Data distribution was investigated using the Kolmogorov–Smirnov test. Data are presented as mean with standard deviations (SD) if not stated otherwise.

Continuous variables were log-transformed if needed to obtain a normal distribution; when the log transformation did not give an approximately normal distribution, nonparametric tests were used. The paired *t*-test or Wilcoxon signed-rank test was used to compare differences in gait parameters between matched cases and controls [42]. 

An index of effect size point biserial correlation, r, is reported for statistically significant differences among groups and between shod and barefoot conditions [43]; effect sizes were defined as 0.1—small, 0.3 medium and 0.5 large [44]. The *p*-value < 0.05 was considered statistically significant. A strong correlation between data of left and right side was found. Data from both sides were used for stride time, stride length and step asymmetry calculations; from right side only loading response, foot contact angle, rearfoot angle and angular velocities were used for statistical analysis. Conditional logistic regression analysis was performed using the COXREG function in SPSS to determine the effect of the statistically significant gait parameters on the risk of lower leg overuse injury. Furthermore, for the statistically significant gait parameters receiver operating characteristic (ROC) analysis was used to examine the area under the curve (AUC), and the specificity, sensitivity, and cut-off value were based on the Youden index [45].

## 3. Results

### 3.1. Gait Parameters

Both groups’ barefoot and shod conditions showed statistically different gait stride characteristics (*p* < 0.001). The barefoot walking showed shorter stride length (r = 0.64) but increased stride time (r = 0.52) and increased stride length variability (r = 0.74) compared to the shod condition in both study groups. Barefoot stride time (*p* = 0.053) and stride time variability (*p* = 0.030) were statistically different between the cases and control group, effect sizes r = 0.31 and r = 0.85, respectively. During the shod walk the stride time was statistically different between the study groups (*p* = 0.048, r = 0.36). Table 2 presents the gait characteristics for barefoot and shod conditions. 

### 3.2. Foot and Ankle Joint Kinematics

Foot and ankle motion analysis during shod and barefoot walking differed in both groups and showed no dissimilarities between cases and control subjects. Foot contact angle increased during the shod walking, but rearfoot eversion angle and angular velocities decreased. See Table 3 for details.

### 3.3. Regression Analysis

In univariate and multivariate analysis, only stride time variability during barefoot gait could significantly predict the risk of lower leg overuse injury. See details in Table 4. Univariate ROC analysis showed an AUC of 0.77 (*p* < 0.001; 95% CI 0.648–0.883), a sensitivity of 56%, and a specificity of 88%, with an optimal cutoff value for barefoot stride time variability of 1.95%.

## 4. Discussion

According to our findings, infantry boots have significant effects on gait parameters, with gait with boots becoming faster, less variable, and more symmetric. Shod gait results support prior research that found military boots design features contributed to body balance [46,47]. The study’s findings on increased stride time and stride length when walking in boots are consistent with earlier research. [24,48]. Furthermore, shod gait analysis shows that military boots decrease ankle joint motion, stabilize the rearfoot and slow the ankle movement during walking, which is consistent with earlier research that investigated barefoot and shod gait during running and walking [24,48,49,50]. Our findings on the maximum angular velocities during ankle dorsiflexion and plantar flexion, as well as range of rearfoot eversion, are comparable with the previously reported data observed in healthy populations [51,52]. Infantry boots usage significantly alters gait parameters, and the evaluation of shod gait can mask the musculoskeletal injury risk of a lower leg. Therefore, barefoot gait assessment protocols can be recommended for the evaluation of military personnel.

The main result of this study is that barefoot stride time variability is significantly related to previous lower leg overuse injuries. The normal range of stride variability in healthy individuals varies from 0.6–2.0% [53], and based on our study results stride variability value among previously injured infantry soldiers is 1.98 ± 0.79. Based on our study findings, a more restricted reference range of stride time variability could be considered in specific physically active populations, such as the armed forces. Furthermore, regression analysis showed that stride time variability is greater than 1.95%, and lower leg overuse injuries can be predicted with 88% specificity and 56% sensitivity. Our prediction should have both high sensitivity (true cases—those who will experience an event) and high specificity (correctly identify true non-cases). However, in practice, there is a trade-off between sensitivity and specificity, and high specificity is more important when screening recruits for a low prevalence outcome. Nevertheless, our finding regarding increased barefoot stride variability is consistent with previous prospective study reporting an association of stride time variability with the musculoskeletal injury risk among Israeli Defense Forces soldiers [40], but the possible cutoff value for the stride time variability has not been set previously. Prospective studies on healthy individuals are needed to evaluate stride time variability cutoff value as a potential lower leg overuse injury risk factor. For possible musculoskeletal injury mitigation, stride variability could be corrected through knee extension and hip abduction strength training or during gait retraining [54,55].

Our study results are limited due to several factors. This study was a case-control study and it could be discussed whether change in barefoot stride time variability is a result of an overuse injury or a protective mechanism. We did not find significant differences in body height or foot sole length between the research groups, so we did not modify the gait data for these characteristics that might shift the results, even though stride duration differences may emerge due to anthropometric factors [56,57]. Gait variability may have been influenced by a history of musculoskeletal overuse injury. Although recovery from injury can vary widely among individuals [58,59], all study subjects were free of any injury, felt healthy, and did not report any symptoms or functional limitations that could influence walking patterns throughout the gait testing. We also tested our study subjects in a gait laboratory, and stride data measured under certain conditions cannot be easily transferred to other conditions [24]. Nevertheless, soldiers during the walking trials used the same infantry boots they use daily and not an experimental pair, which could lead to a more natural gait pattern. Additionally, our study results cannot be generalized to all types of shoes worn in the military, because soldiers also use running shoes during service time. Only one infantry boot type was used and we did not investigate different military boot features that could impact the result. For example, Helton et al. found that running shoes with mild to moderate lateral-torsional stiffness were effective in reducing the lower extremity injury risk among military cadets [60]. 

Additionally, we did not analyze shoe attrition, but Chen et al. recently postulated that running shoe attrition impacts the kinematics and kinetics of lower extremity joints [36], and we do not know if it is the same for the infantry boot. Footwear in the Latvian Land Forces is changed regularly if visible shoe attrition persists, and no visual damage (e.g., asymmetrical shoe heel abrasion) of the infantry boots was found before marker placement during the study period. 

Moreover, the rearfoot and ankle joint motion tracking with markers during barefoot gait analysis can be a source of error due to soft tissue artifacts (STA); however, the STA in the heel is likely to be small [61,62], but STA could influence ankle joint motion results. The marker set for the foot motion analysis, as well as marker placement errors, might shift the results. However, during the study, all markers were placed by one examiner following the standardized scheme of marker placement. The rearfoot motion findings are also consistent with previous study results with a similar marker set (11 markers, without a second metatarsal head marker) [63]. 

For shod analysis, we have used shoe-mounted markers that do not fully represent foot motion [64]. Other study findings obtained from shoes with holes in the heel have reported differences from the findings of shoes with an intact heel, but high accuracy of the placement of the shoe marker was reported for the hindfoot and forefoot [61,65]. Additionally, infantry boots with holes could not be used by soldiers afterward and would need to be replaced, which would have increased the study expenses and caused inconvenience for the study participants. 

Despite these limitations, this study adds knowledge to gait-related parameters in military personnel in terms of lower leg overuse injuries. To the author’s knowledge, this is the first case-control study to evaluate gait parameters as possible risk factors for lower leg overuse injuries in infantry soldiers. The findings of our study emphasize the importance of gait variability as a possible lower leg overuse injury risk factor among infantry soldiers, and gait analysis can be considered for screening and training purposes. 

## 5. Conclusions

Overuse injury risk is independent of stride-related characteristics during walking in infantry boots. Shod gait analysis may underestimate the risk of a lower leg overuse injury because military boots modify gait parameters. A stride time variability of more than 1.95% during barefoot walking is the strongest predictor of lower leg overuse injury in infantry soldiers. In the military, increased gait variability should be considered as a possible predictive factor for lower extremity overuse injury.

## Figures and Tables

**Table 1 ijerph-20-06449-t001:** Characteristics of study subjects (Mean ± SD).

	Case (*n* = 32)	Control (*n* = 32)	*p*-Value
Age, years ^1^	29.13 ± 5.77	30.78 ± 5.13	0.087
Height, m	1.81 ± 0.08	1.77 ±0.07	0.103
Weight, kg	81.09 ± 13.54	81.75 ±12.53	0.731
BMI	24.74 ± 2.90	25.94 ± 2.85	0.100
Foot sole length, mm	275 ± 12	272 ± 12	0.488

^1^ SD—standard deviation, BMI—body mass index, mm—millimeters.

**Table 2 ijerph-20-06449-t002:** Gait characteristics of case and control groups (Mean ± SD).

	Case	Control	*p*
Barefoot			
**Stride time, SD**	**1.11 ± 0.09**	**1.04 ± 0.12**	**0.053 ***
**Stride variability, %**	**1.98 ± 0.79**	**1.27 ± 0.66**	**0.030**
Loading response, %	12.11 ± 2.26	12.12 ± 2.04	0.962
Step length asymmetry index	0.56 ± 5.55	0.42 ± 3.74	0.893
Stride length, m	1.14 ± 0.32	1.08 ± 0.33	0.176
Stride length variability, %	1.88 ± 1.72	1.97 ± 1.88	0.165
Shod			
**Stride time, SD**	**1.24 ± 0.01**	**1.19 ± 0.09**	**0.048 ***
Stride variability, %	1.24 ± 0.85	1.21 ± 0.73	0.629
Loading response, %	11.83 ± 2.45	10.69 ± 1.53	0.132
Step length asymmetry index	0.53 ± 4.56	0.12 ± 1.03	0.332
Stride length	1.34 ± 0.26	1.32 ± 0.30	0.571
Stride length variability, %	0.81 ± 0.73	0.72 ± 0.63	0.630

* Significant results marked in bold; SD—standard deviation.

**Table 3 ijerph-20-06449-t003:** Foot and ankle complex movements with standard deviations during barefoot and shod gait.

Barefoot	Group		
	Case	Control	*p*
Foot contact angle (°)	16.41 ± 5.86	17.04 ± 5.18	0.487
Rearfoot eversion (°)	5.64 ±1.96	4.97 ± 1.65	0.692
Peak angular velocity, PF ^1^ (°/s	242.17 ± 36.71	256.4 ± 30.17	0.138
Peak angular velocity, DF ^1^ (°/s)	157.38 ± 28.62	149.52 ± 14.04	0.201
Shod	Case	Control	*p*
Foot contact angle (°)	25.31 ±4.77	25.38 ± 4.63	0.896
Rearfoot eversion (°)	3.28 ±1.10	2.88 ± 1.11	0.147
Peak angular velocity, PF (°/s)	157.47 ± 23.99	162.32 ± 26.79	0.475
Peak angular velocity, DF (°/s)	119.14 ± 36.36	120.07 ± 30.69	0.915

^1^ PF—plantarflexion, DF—dorsiflexion, s—seconds.

**Table 4 ijerph-20-06449-t004:** Summary of conditional logistic regression analysis.

	Barefoot		Shod	
Variable	Unadjusted OR (95% CI)	Adjusted OR (95% CI)	Unadjusted OR (95% CI)	Adjusted OR ^1^ (95% CI)
**Stride time** **variability**	**2.59** (1.30–5.18)	**2.71** (1.31–5.60)	1.01 (0.99 –1.01)	1.00 (0.97–1.04)
*p*	**0.009 ***	**0.007 ***	0.928	0.131

^1^ OR—odds ratio; CI—confidence interval; * significant results marked in bold.

## Data Availability

The Latvian National Army Logistics Command Military Medical Support Centre did not allow data sharing, and the analyzed data sets are not publicly available. Requests to access the data sets should be directed to the corresponding author.

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
