# Peer review of "Increased Barefoot Stride Variability Might Be Predictor Rather than Risk Factor for Overuse Injury in the Military"

_ijerph, 2023, doi:10.3390/ijerph20156449_

Round 1

Reviewer 1 Report

Dear Authors,

It's been an honour to review your paper "Increased stride time variability as a possible risk factor of the lower extremity overuse injury in the military".

You had a good introduction; however, I suggest clarifying the study's aim in lines 51 to 54.

There is a repetition in lines 63-66 of lines 59-60.

It would be best to have a picture of the experiment settings and a figure to explain precisely what you measured, gait cycles and the instrumentation.

I find the presentation of results appropriate.

I agree that the footwear modifies the movement pattern, including the stride time variability.  

Still, I have some concerns with the study design and the discussion. I can't see how you can create a predictor of injury based on the data collected from recovered cases. Please clarify how the completeness of recovery from the previous injury relates to future injuries and how this correlation can be used as evidence.

To compare the stride times of your cases and control group, I suspect the effect of proportional anthropometric data. So it's great to know the height and the BMI index, but the length of the sole (or the footwear size), the hip height, and the length of the lower and upper leg are independent variables that can have a significant effect too.

Finally, please explain why you chose a 6-month healing period to select your cases.

Author Response

Thank you for your comments. 

Changes are highlighted in yellow in the manuscript text. Please find all the corrections in attachment.

Reviewer 2 Report

Is it possible to differentiate whether the stride time variability was increasing or decreasing?

There is no description of the physical parameters of the soldiers: weight, height, number of feet, ...

Is it possible to analyze the data by groups/ranges of weight, height, ...?

Correct the following:

Line 263: add doi: 10.1097/JSA.0000000000000246.

Line 268: add doi: 10.1016/j.jsams.2021.03.016.

Line 276: add doi: 10.1016/j.amepre.2020.08.007.

Line 293: change doi:https://doi.org/... to doi: 10.1016/j.gaitpost.2006.01.002

line 305: change doi:https://doi.org/... to doi: 10.1016/j.gaitpost.2021.04.024

line 331: delete acessed on 10 february 2019 or cite a more recent date

line 335: add doi: 10.1111/j.1469-185X.2007.00027.x

Author Response

Thank you for your comments. Please find the file attached to see all the corrections. 

Reviewer 3 Report

In this study, the authors attempted to investigate the effect of masking or supplementing boots in soldiers with overuse injuries. These ideas sound pretty interesting. However, as a scientific study, authors need more precise detailed descriptions and analyzes to approach and define these topics. Therefore, it is considered that this study can be accepted only when corrected through major revision.

Title: Does this title adequately represent the results of this study?

Introduction

The authors reported the results of masking the variability found in the stride when wearing a boot, unlike barefoot, in a patient with an overuse injury. The introduction should be written according to the flow of this research. However, in the middle, it is full of unrelated sentences including Parkinson's disease.

M&M

This part is the most important part to be modified in this study. The authors have been very confused in their description of this part.

First of all, the authors should accurately indicate the information including more factors related with gait and underlying condition which means overuse injuries of the two groups compared through the table 1.

And it was expressed as an overuse injury, but this criterion is absolutely unacceptable. What is overuse injury defined by the authors? We don't know exactly what the standard is, and we don't know what kind of disease it includes. This is the critical error in this study. There are so many factors that affect gait. If these parts are not precisely defined and controlled, this study is only a comparison of patients by the authors with inaccurate criteria for soldiers.

Result.

The authors found that the stride variability was corrected when wearing the boot. What does stride variability mean? Doesn't each person's walking habits also have an effect?

Discussion

After all, soldiers do not work barefoot, but work in military boots. The authors have discovered the effects of these boots, discuss the effects of boots that lead to such results, and discuss possible theoretical backgrounds that can be found in other studies. Table 4 should be placed in result section. 

Conclusion 

Does the result of this study mean that the stride variability seen in the barefoot gait analysis is a predictor of overuse injury? I think the logical conclusion that can be drawn from this study is that wearing boots in soldiers with lower extremity discomfort has a correction effect to some extent in the gait pattern. This conclusion and the title of the study should be rewritten according to the logic shown by the results of this study.

Overall English proofreading or correction is required. 

Author Response

Thank you for your comments. Please see the attached file for the corrections made. 

Round 2

Reviewer 1 Report

Thanks for the improvement.

Author Response

Thank you for your comments.  We appreciate your interest in our work.

Reviewer 3 Report

A lot of things have been modified, but there are still a few things that need to be improved.

Title: 

Increased barefoot stride variability might be predictor rather than risk factor for overuse injury.

Abstract

line 18: please add SD or SE. 

line 19: same with title. is it predictor or risk factor? 

introduction

line 55: Same point as the title. Isn't it the conclusion of this study that if stride variability is observed, we can expect the soldier to have an overuse injury of the lower extremity?

Table 2, 4: please add "*" for significant p-values.

line 250-251: it's same. please reconsider the expression, "risk factor".

Acceptable. 

Author Response

Dear Reviewer, 

We appreciate your interest in our work. 

We have changed the title.  Made corrections to the 'Abstract' section, line 18 - we added CI based on the ROC curve analysis results. 

Corrections also were made to the 'Abstract section', line 19; 'Introduction' section, line 55.

Additionally, "*" were added for significant p-values in the Table 2 (line 148) and Table 4 (line 161). 

We also changed the expression "risk factor" to 'predictive factor', line 251.

Thank you for your advice.